Original research

# Using community-based, participatory qualitative research to identify determinants of routine vaccination drop-out for children under 2 in Lilongwe and Mzimba North Districts, Malawi

Jocelyn Powelson [1], Joan Kalepa,[2] Hannah Kachule,[2] Katie Nkhonjera,[2] Charles Matemba,[2] Mike Chisema,[3] Tuweni Chumachapera,[3] Emily Lawrence[1]

JP and JK contributed equally.

¹VillageReach, Seattle, Washington, USA
²VillageReach, Lilongwe, Malawi
³Malawi Ministry of Health, Lilongwe, Malawi

**Correspondence to**
Jocelyn Powelson;
jocelyn.powelson@villagereach.org

## ABSTRACT

**Objective** In recent years, full childhood routine immunisation coverage has fallen by 5% to levels not seen since 2008; between 2019 and 2021, 67 million children were undervaccinated. We aimed to identify and describe the determinants of vaccination drop-out from the perspectives of caregivers and health workers in Malawi.

**Design** We used a community-based participatory research approach to collect data through photo elicitation, short message service exchanges, in-depth interviews and observations. We used a team-based approach for thematic analysis, guided by the Behavioural and Social Drivers of Vaccination framework.

**Setting** The study was conducted in Lilongwe and Mzimba North Districts in Malawi, representing urban and rural settings, respectively.

**Participants** Participants included caregivers of partially vaccinated (n=38) and fully vaccinated (n=12) children between 25 and 34 months and Community Health Workers (n=20) who deliver vaccines. Caregiver participants were identified through health facility vaccination registers and with the assistance of community health volunteers.

**Results** We identified five principal drivers of routine vaccination drop-out: (1) poor caregiver knowledge of the vaccine schedule and how many vaccines are needed for full vaccination; (2) caregivers' fear of repercussions after not following vaccination guidelines; (3) rumours and concerns if vaccines are repeated or new ones are introduced; (4) high opportunity cost of health facility visits, exacerbated by wait times, stockouts and missed opportunities and (5) limited family support and vaccination burden placed largely on mothers. Key differences between rural and urban settings related to practices around health cards and vaccine wastage, wait times, migrant and tenant communities, and social support systems.

**Conclusions** Immunisation interventions should be tailored to address drivers of drop-out in the community, the health facility and beyond. Service quality, timeliness and reliability need to be improved, and tailored messaging

## STRENGTHS AND LIMITATIONS OF THIS STUDY

⇒ Photo elicitation methods allowed participants to visually describe their experiences, resulting in rich, detailed and nuanced findings on barriers to complete immunisation.

⇒ Caregivers from the study communities were highly involved in both data collection and analysis, ensuring that community perspectives were integrated into the findings and resulting recommendations.

⇒ During the data collection period, there were multiple vaccination campaigns (for polio and cholera vaccines) across Malawi, as well as pilots of the malaria vaccine in some of the study sites, which may have introduced recall bias and influenced some of the findings that related to caregivers' perceptions of campaigns and new vaccine introduction.

⇒ There may have been selection bias resulting from (1) inconsistent participant recruitment methods between the study sites due to differences in vaccination registry practices and (2) difficulties in recruiting participants from migrant or tenant communities as well as certain religious communities due to those caregivers moving locations during the recruitment period or refusing to participate.

and education are needed, especially in response to COVID-19-related misinformation and introductions of new, routine vaccines.

## INTRODUCTION

Between 2019 and 2021, an estimated 67 million children did not complete the routine childhood immunisation sequence that protects them from life-threatening diseases.[1] By 2021, global immunisation coverage for infants had fallen to 81%, the lowest rate in over a decade, largely due to the COVID-19 pandemic's crippling impact on global

vaccine supply chains and disruptions to routine health services.[2]

At the start of this study, Malawi's National Immunisation Programme recommended that all children receive 16 doses of vaccines before age 2, including 1 dose of BCG, 3 of pentavalent vaccine (DPT-HepB-Hib), 4 of oral polio vaccine (OPV), 1 of inactivated polio vaccine, 3 of pneumococcal conjugate vaccine (PCV), 2 of rotavirus vaccine and 2 of measles containing vaccines (online supplemental appendix A).[3] The country is in the process of introducing the typhoid conjugate vaccine and four doses of the malaria vaccine into the routine schedule.[3 4] Vaccines are delivered both through the public sector, including at static facilities, outreach sessions in communities and large-scale vaccination campaigns, as well as through the private sector, including in for-profit and non-profit facilities. The government purchases and freely provides vaccines, but caregivers who seek services from private facilities often pay fees for children to be weighed prior to vaccination.[5] A cadre of community health workers called Health Surveillance Assistants (HSAs) are responsible for delivering vaccines at all sites. HSAs also perform other tasks within communities related to health education and sanitation and hygiene.[6 7] Though outside of the formal health system, within many communities, Care Groups composed of volunteers and mothers play an important role in spreading health information and mobilising communities.[8 9]

Addressing barriers to full vaccination is especially important in Malawi, where WHO and UNICEF estimates of coverage have dropped during the last decade for most vaccine doses; for example, coverage estimates for BCG, DPT-HepB-Hib 3, OPV 3 and PCV 3 were all 95% or higher in 2012, but then fell to 86% or lower by 2016.[10] While routine vaccination coverage increased between 2016 and 2019, coverage for almost all doses on the routine schedule declined by approximately 15 percentage points between 2019 and 2022, in large part due to the COVID-19 pandemic.[10] Coverages for most doses were between 82% and 89% in 2022; however, measles second dose coverage continues to lag behind, at an estimated 60% in 2022.[10] Contrary to similar settings in sub-Saharan Africa, Malawi's coverage rates are higher in rural areas than in urban areas, at 77% and 70% full vaccination coverage, respectively, in 2015–2016.[11] Reviews of reporting forms and anecdotal evidence from the facilities involved in this study indicate that vaccination record-keeping tools and processes also differ between urban and rural facilities; in rural regions such as Mzimba North, facilities track individual children's records in paper-based vaccination registers, while in urban settings such as Lilongwe, facilities keep daily, paper-based tallies of antigens delivered, but do not keep records for individual children due to the much higher patient numbers as well as the likelihood of caregivers visiting multiple different facilities for vaccination services.

Previous research in low-income and middle-income countries has identified barriers at the individual, interpersonal and health systems levels that can lead to vaccination drop-out, including poor access to facilities, unreliable and poorly perceived service quality, fear of side effects, lack of family support, gender dynamics, health worker availability, missed opportunities, childcare challenges for siblings, lack of motivation, poor attitudes and behaviour by health workers, loss of vaccination cards and displacement or migration of caregivers.[12–19] In Malawi in particular, previous studies have identified additional barriers including poor understandings of vaccination schedules, rumours or misconceptions fueled by religious beliefs, insufficient communication between health workers and communities, and unreliable outreach services.[20 21] Given Malawi's unique situation as a country where vaccination coverage has generally been declining in the last decade and where urban coverage is lower than rural, there is a need for more evidence on determinants of vaccination drop-out in both rural and urban settings in Malawi. We aimed to identify and describe the determinants of vaccination drop-out from the perspectives of caregivers and health workers.

## METHODS
### Study design
We conducted a qualitative study using a community-based participatory research (CBPR) approach.[22] CBPR is an approach that engages community representatives throughout the research process, helping to minimise bias and reduce power dynamics during data collection and facilitating development of contextually sensitive and community-centred findings.[23] We hired and trained four caregivers from the study communities (hereafter referred to as 'caregiver researchers') to lead data collection and assist with analysis.

Caregiver researchers were recruited from the study districts via an open job posting. Applicants were required to have a child 36 months or younger, and women were prioritised as they most closely represent the study population. Two women in each district who met those requirements were selected as caregiver researchers based on their familiarity with the study communities, fluency in local languages and in English and their prior research experience. All four caregiver researchers had bachelor's degrees and basic data collection and research skills and had recently experienced the under-2 vaccination journey with their own children. They participated in a 5-day training on qualitative methods and research ethics that covered interview techniques such as how to minimise bias and how to probe for rich, detailed information.

We adapted the WHO's Behavioural and Social Drivers (BeSD) of vaccination framework,[24] which is an established framework for qualitative research on caregivers' immunisation experiences. We added components from UNICEF's immunisation journey by breaking down the 'practical issues' domain into practical issues during preparation, point-of-service and after care to enable development of more nuanced themes.[25] This model

served as the framework to guide data collection and analysis (online supplemental appendix B).

## Study setting, population and recruitment

The study was conducted between July 2022 and February 2023 in Lilongwe and Mzimba North Districts in the Central and Northern Regions of Malawi, respectively. According to the Malawi Harmonised Health Facility Assessment conducted in 2018–2019, there were 115 health facilities (including health centres, health posts, clinics, hospitals and dispensaries) in Lilongwe serving a population of over 2.6 million, and 42 facilities in Mzimba North serving a population of roughly 940 000. At that time, Lilongwe had 6.7 health staff for every 10 000 people, compared with Mzimba North with 15 per 10 000.[26 27]

To select study sites, we worked with the Malawi Expanded Programme on Immunisation (EPI) who provided us with a list of health facilities with the highest drop-out rates. We then worked in coordination with the national and district EPI to purposively select eight of those facilities (four per district) that represented both rural and urban sites and that were fairly accessible. We collected data at both static and outreach sites for each health centre. The study population included caregivers in the catchment areas of the selected health centres as well as HSAs responsible for delivering vaccines at those sites. 'Caregivers' were defined as parents or guardians who took primary healthcare-seeking responsibility for a child. We included both caregivers of fully vaccinated children (hereafter referred to as 'FV caregivers') and caregivers of partially vaccinated children (hereafter referred to as 'PV caregivers'). Caregivers were eligible if their child was 25–34 months old, an age by which they should have completed the under-2 immunisations while still being early enough to minimise recall bias. Vaccination status was determined by whether or not the child had completed all 16 doses on the routine immunisation schedule by age 2.

In Mzimba North, eligible caregivers were identified through health facility vaccination registers. In Lilongwe, where individual children's vaccinations are not tracked at the health facility, Community Health Volunteers identified eligible caregivers. In both districts, caregiver researchers used a convenience sampling approach to visit eligible caregivers' homes until they reached enrolment targets for PV and FV caregivers in all sites. They confirmed eligibility and vaccination status through children's health cards, when available. Caregivers in Lilongwe who were missing health cards were asked to describe which vaccines their children had received and were excluded if they were uncertain of their child's vaccination status. Two PV caregivers went to the facility to complete their children's missing vaccinations immediately after the recruitment visit. They were excluded from the study due to concerns that they went to complete vaccination out of fear of being reprimanded and that this fear might limit their comfort to provide genuine, unbiased responses in the interviews. Twenty-six caregivers in Lilongwe and 24 caregivers in Mzimba North were enrolled, at which point data saturation was met (figure 1). The study team purposively recruited and enrolled 20 HSAs from both static and outreach sites across all 8 facilities. No HSAs declined to participate.

## Data collection

During recruitment, caregiver researchers explained the study, collected demographic information and obtained informed consent with participants' signatures or thumbprints. Tools were each piloted once and refined prior to data collection. Data were collected in Tumbuka, Chichewa or English depending on participants' preferences. Four different methods were used to collect data from caregiver and HSA participants in the following order: 20 PV and 6 FV caregivers from Lilongwe and 18 PV and 6 FV caregivers from Mzimba North participated in photo-elicitation interviews; 12 HSAs from Lilongwe and 8 from Mzimba North participated in message exchanges; observations were conducted at 4 static and 4 outreach sessions in each of Lilongwe and Mzimba North; and 11 HSAs from Lilongwe and 8 from Mzimba North participated in in-depth interviews (figure 2).

Photo-elicitation interviews were conducted using semi-structured guides based on the BeSD framework (online supplemental appendix C). Photo-elicitation is a method through which photos are incorporated into the interview process, allowing participants to engage with and react to visual material to generate rich, detailed findings[28]; for

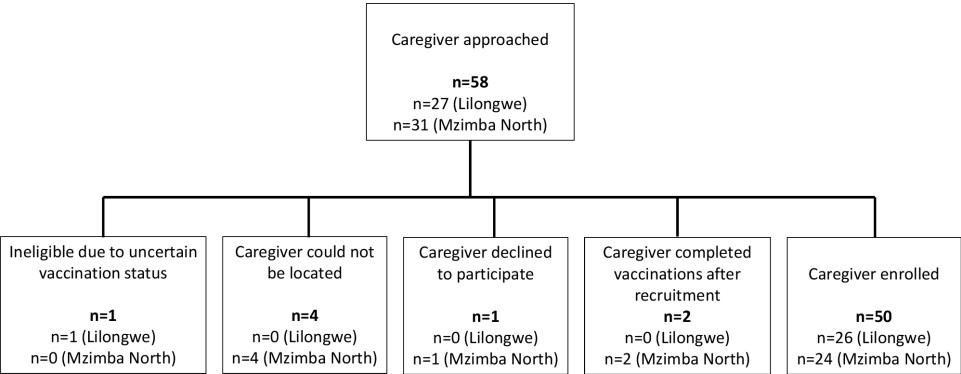

**Figure 1** Recruitment and enrolment of caregiver participants.

| | Step 1:<br>Photo-Elicitation Interviews | Step 2:<br>Message Exchanges on Telegram | Step 3:<br>Observations of Immunization Sessions | Step 4:<br>Interviews with Health Surveillance Assistants (HSAs) |
|---|---|---|---|---|
| Participants | • 26 caregivers from Lilongwe<br>• 24 caregivers from Mzimba North | • 12 HSAs* from Lilongwe<br>• 8 HSAs from Mzimba North | • 4 static and 4 outreach sessions in Lilongwe<br>• 4 static and 4 outreach sessions in Mzimba North | • 11 HSAs from Lilongwe<br>• 8 HSAs from Mzimba North |
| Time-frame | July – September, 2022 | 3 weeks each between Oct '22 – Jan '23 | 4 hours per observation between Dec '22 – Feb '23 | Following observations, between Dec '22 – Feb '23 |
| Description | Semi-structured interviews leveraging photos to generate rich data on caregiver perspectives on vaccination journeys | Message exchanges discussing HSA's observations of notable events or situations arising during immunization sessions | Observations of the environment and settings, general processes, flow of patients, and interactions between caregivers and HSAs at immunization sessions | Semi-structured interviews that generated data on HSAs' experiences with immunization and their perspectives on causes of dropout |

**Figure 2** Data collection process and participants. *HSAs, Health Surveillance Assistants.

example, participants might be shown a set of photos and then asked to describe if and how those photos relate to their own lives. These methods built on a sister study on vaccination drop-out in Mozambique that employed photovoice methods, through which caregivers were provided with cameras and instructed to take photos of anything that represented their experiences vaccinating their children, resulting in a set of 159 photos of different aspects of the immunisation journey taken between October and November 2020.[12] The lead research team selected 37 of these photos that depicted the variety of vaccination barriers and facilitators that arose during that study to serve as potential material for the photo-elicitation methods. The caregiver researchers reviewed these photos and took additional photos of country-specific factors in Malawi, such as an HSA in the Malawi uniform. They then selected a final set of 29 photos for the photo-elicitation interviews, including 24 from Mozambique and 5 from Malawi, that they felt encompassed a wide range of potential drop-out determinants across the immunisation journey (online supplemental appendix D). Informed consent was previously obtained for use of all selected photos.[12]

At the start of the interviews, caregivers were asked to reflect on their immunisation visits and to think about photos or depictions that might represent their immunisation journey. Participants then examined the set of 29 photos and selected the 5 that they felt best represented their vaccination experiences. Caregiver researchers discussed the selected photos with the participants to learn more about how they related to their experiences before continuing with the semistructured interview questions. EL and JP reviewed each caregiver researcher's first transcript and provided feedback to improve the quality of subsequent interviews. The full research team (JP, JK, EL, KN, HK, GN and CM) met weekly to debrief on interviews and discuss areas for further probing.

HSAs participated in message exchanges and semi-structured interviews in Tumbuka, Chichewa or English to share their experiences delivering childhood immunisations and their perspectives on vaccination drop-out.

During recruitment, the caregiver researchers provided basic smartphones to HSAs who did not have their own, installed the Telegram messaging application and oriented HSA participants on how to use it. The caregiver researchers prompted HSAs to send messages or photos whenever they had a notable experience relating to under-2 immunisations. The caregiver researchers replied to messages to learn more details and sent weekly follow-up reminders. Individual message exchanges lasted 3 weeks and took place between October 2022 and January 2023. The HSAs sent a median of 26.5 messages (range: 1–74) during the 3 weeks. Network challenges limited some HSAs from sending or receiving messages regularly, hence the wide range in the number of messages sent.

After completing the interviews with caregivers and the message exchanges, the caregiver researchers conducted 16 observations of immunisation sessions, including at both static and outreach sites for all 8 facilities. Observation sessions lasted approximately 4 hours, during which the caregiver researchers took notes on the environment in which the session was conducted, the general flow of the sessions, attitudes and interactions between caregivers and health workers, and what kind of immunisation education was presented (online supplemental appendix E). Following these sessions, the caregiver researchers conducted semistructured interviews with 19 of the enrolled HSAs; 1 HSA dropped from the study prior to the semistructured interviews due to conflicting schedules. Interview guides were based on the BeSD framework, and caregiver researchers asked additional questions based on notable points that arose during the message exchanges and observations (online supplemental appendix F).

## Data analysis

Audiorecordings of interviews were transcribed and translated into English by the caregiver researchers. The initial codebook was based on a codebook developed during a similar study on vaccination drop-out in Mozambique that used the same adapted BeSD framework; that codebook was developed deductively based on the BeSD constructs

and UNICEF immunisation journey. The codebook was updated with new codes added inductively by JP and EL following review of the first six caregivers' interview transcripts and was presented to the caregiver researchers for additional edits in order to tailor the codebook to the Malawi context. JP and EL individually coded the first two caregivers' interview transcripts in ATLAS.ti V.23 and reviewed the coded transcripts to establish intercoder agreement. JP and EL divided and independently coded the remaining caregiver interview transcripts. JP, EL, JK, HK, KN and GN summarised the main findings by vaccination status and district. Caregiver researchers participated in a 1-week participatory analysis process facilitated by JP, in which they reviewed code summaries and representative quotations from the caregiver interviews, discussed key facilitators and barriers to full vaccination within each category of the adapted BeSD framework, identified themes and patterns of drop-out that fell across the BeSD framework categories, and drew comparisons between PV and FV caregivers and between Lilongwe and Mzimba North. Photos selections were analysed to compare photo selection of FV versus PV caregivers and of Lilongwe versus Mzimba North caregivers.

On review of the first set of transcripts from HSA message exchanges, interviews and observations, additional codes were added inductively to the codebook to capture HSAs' perspectives. JP, EL and NM coded HSA transcripts and observations, and JP and NM summarised findings. The themes generated from the caregiver data were then updated to reflect additional insights from the HSA data and were reviewed and validated by the caregiver researchers before being finalised.

### Patient and public involvement
Four caregiver researchers, as representatives of the study population, participated in data collection and analysis. Findings were shared with many participants, HSAs and with local health authorities during follow-up workshops in February and March 2023.

## RESULTS
### Participants
Fifty caregivers participated in the study, including 47 mothers, 1 aunt, 1 grandmother and 1 father. All FV caregivers were mothers of the children. In general, FV caregivers were more educated than PV caregivers. PV children were much more likely to be female compared with FV children. PV children were missing an average of three doses (range 1–14 missing doses) (table 1). Measles-rubella (MR) 2 had the lowest uptake among PV children, followed by MR 1, PCV 3 and OPV 3 (table 2).

Twenty HSAs participated in the study (table 1). Most had completed secondary education and had been working as HSAs for more than a decade. All but one had children of their own, though none had any children under 2 years. All HSAs had received some form of immunisation-related training in the past 6 months, mostly associated with the national polio campaigns.

### Key influences on routine vaccination drop-out
We identified barriers across all domains of the BeSD Model. In most cases, caregivers dropped out after encountering multiple barriers across different categories. Our analysis of data from caregivers, HSAs and

| Table 1 | Characteristics of caregiver participants and their children and of HSAs | | | |
|---|---|---|---|---|
| **Characteristic** | **Category** | **PV group (n=38)** N (%) or median (IQR) | **FV group (n=12)** N (%) or median (IQR) | **HSAs (n=20)** N (%) or median (IQR) |
| District | Lilongwe | 20 (53) | 6 (50) | 12 (60) |
| | Mzimba North | 18 (47) | 6 (50) | 8 (40) |
| Caregiver/HSA age (years) | | 28 (25–34) | 25 (23.5–25.3) | 36.8 (42.5–49.3) |
| Caregiver/HSA education | None | 3 (8) | 0 | 0 |
| | Some primary | 11 (29) | 5 (42) | 0 |
| | Completed primary | 17 (45) | 3 (25) | 0 |
| | Some secondary | 5 (13) | 1 (8) | 4 (20) |
| | Completed secondary | 2 (5) | 3 (25) | 14 (70) |
| | Some tertiary | 0 | 0 | 1 (5) |
| | Completed tertiary | 0 | 0 | 1 (5) |
| Child age (months) | | 30 (26–31) | 28 (26.8–29.3) | – |
| Child/HSA sex | Male | 15 (39) | 9 (75) | 7 (35) |
| | Female | 23 (61) | 3 (25) | 13 (65) |
| # of missing doses (of 16 possible) | | 1 (1–4.8) | 0 | – |
| Work experience as HSAs (years) | | – | – | 14.5 (9.5–20.3) |

FV, fully vaccinated; HSA, Health Surveillance Assistant; PV, partially vaccinated.

Table 2 Vaccination completion of PV children of caregiver participants

| Vaccine | Dose | No (%) of PV children who received the dose (n=38) |
|---|---|---|
| Tuberculosis | BCG | 38 (100) |
| Oral Polio | OPV 0 | 36 (95) |
| | OPV 1 | 35 (92) |
| | OPV 2 | 32 (84) |
| | OPV 3 | 27 (71) |
| Inactivated Polio | IPV | 29 (76) |
| Pentavalent: diphtheria, pertussis, tetanus, hepatitis B, haemophilus influenzae type B | DPT-HepB-Hib 1 | 37 (97) |
| | DPT-HepB-Hib 2 | 32 (84) |
| | DPT-HepB-Hib 3 | 29 (76) |
| Pneumococcal conjugate | PCV 1 | 35 (92) |
| | PCV 2 | 31 (82) |
| | PCV 3 | 27 (71) |
| Rotavirus | RV 1 | 35 (92) |
| | RV 2 | 30 (79) |
| MR | MR 1 | 25 (66) |
| | MR 2 | 8 (21) |

IPV, inactivated polio vaccine; MR, measles-rubella; OPV, oral polio vaccine; PCV, pneumococcal conjugate vaccine; PV, partially vaccinated.

observations revealed five main patterns of barriers that lead to routine immunisation drop-out, including (1) poor caregiver knowledge of the vaccine schedule and how many vaccines are needed for full protection; (2) caregivers' fear of repercussions after not following vaccination guidelines; (3) rumours and concerns if vaccines are repeated or new ones, including for COVID-19, are introduced; (4) opportunity cost of each trip to the facility, exacerbated by long wait times, stockouts and other missed opportunities and (5) limited family support and vaccination burden placed largely on the mother.

### Poor caregiver knowledge of the vaccine schedule and how many vaccines are needed for full protection

Caregiver participants shared that they learnt about the importance of vaccination from many sources, including the facility, radio, community leaders and during campaign events. Consequently, most caregivers in our study understood the importance of vaccines for preventing disease and wanted their children to complete the routine vaccinations. However, not all caregivers were as knowledgeable about other aspects of vaccination, including which vaccines should be delivered when, and how many are needed to complete the sequence.

Caregivers explained that when they went for vaccinations, they might arrive after the health talk was delivered,

or said that the HSAs sometimes skipped the health talk entirely. Furthermore, some caregivers shared that they felt uncomfortable speaking up if they had questions.

> I would be late [for the health talk] and I would find that they had already started administering the vaccines… I forgot the date and month that they were supposed to get that vaccine, and by the time I remembered, months had already gone by.—PV caregiver from Mzimba North

Several HSA participants acknowledged that they do not always deliver health talks or provide adequate information to caregivers. One challenge they noted was that they prefer to wait for enough caregivers to arrive before delivering health talks, but oftentimes, caregivers arrive late, and by the time enough caregivers have arrived, the HSAs are already busy delivering vaccinations. In addition, several HSAs in Lilongwe felt that caregivers are too impatient for a health talk and prefer that the immunisations begin as quickly as possible. The HSAs noted that when facilities are busy, which is often the case in Lilongwe, they do not have enough time to speak individually with caregivers.

> They did not explain. If they did, I would have followed the right procedure because they would tell me to go in such a month on such a date.—PV caregiver from Lilongwe

Caregivers and HSAs described these challenges resulting in caregivers having poor knowledge of the vaccine schedule or of how many vaccines are needed for full protection, which can cause them to drop out early. During recruitment, a number of caregivers believed that their children had completed their vaccinations and were unaware of the second dose of measles vaccine. During observation sessions, caregiver researchers observed that in several facilities, the posters showing the vaccination schedule were outdated and did not include measles 2, which was added to the routine schedule in 2015.[29] In some cases, caregivers were aware that their children were missing one or two vaccines, but they felt that their children were already protected by their other vaccines.

> I am very much satisfied because my child received most of his vaccines. So for me, it was okay even if he couldn't get the last vaccine, which is measles 2.—PV caregiver from Mzimba North

### Caregivers' fear of repercussions after not following vaccination guidelines

Many PV caregiver participants perceived that vaccination, family planning and facility-based deliveries were mandatory and that they would be reprimanded or face other consequences if they did not abide by these guidelines.

> On vaccines… we are in the hands of the government, so for us to stop vaccinating our children it will

be breaking the law which is not good.—PV caregiver from Mzimba North

Caregivers' fear of repercussions may originate in part from certain practices that are enforced at the community or health facility levels. Several caregivers and HSAs mentioned that certain community leaders enforced strict practices around vaccination and facility-based births.

There is one village head who put up policies in his village that any pregnant woman is supposed to deliver at the hospital and not at home and that every woman must go with their children for vaccination. If they don't do that then there will be consequences.—HSA from Mzimba North

At the facility level, several caregivers mentioned that if a sick child is brought to the facility for treatment of an illness, the HSAs first check the health card and require the child to receive missing vaccines before being treated. Some caregivers even associated vaccination status with national identity, saying that the BCG scar on the arm was a sign of being Malawian.

While this perception of vaccination being mandatory can serve as a motivator to stay on schedule, it can also result in caregivers fearing they will be yelled at or accused of negligence by HSAs if a child has fallen off-schedule, they have inadequate birth spacing, or they had a home childbirth. Even when caregivers understood that vaccination is not mandatory, they still feared repercussions for not following these guidelines. In these cases, caregivers may decide not to return to the facility to avoid being reprimanded.

I was also afraid of going to the hospital, [not knowing] whether I could bring my child to take the vaccine that he had skipped. I was so afraid that I just gave up on that vaccine.—PV caregiver from Lilongwe

We health workers should not scold clients but advise them with love… The woman came late to start the scale because she was afraid of our speech [sic].—HSA from Lilongwe

Caregivers in Lilongwe were also hesitant to return to the facility after losing or damaging the child's health card. Health cards are especially critical for tracking vaccinations in Lilongwe, where facilities do not keep records for individual children. In Mzimba North, however, facilities can look up a child's vaccination record if the card is lost, making caregivers less fearful of returning without a card.

My child did not complete the last vaccine. I lost the book, but even when I saw the doctors, I was not asking [about the vaccine]. I was so afraid that if I asked, the person was going to shout at me or even insult me a lot.—PV caregiver from Lilongwe

## Rumours and concerns if vaccines are repeated or new ones, including for COVID-19, are introduced
### Subtheme 1: repeat vaccination
Many caregiver participants said that their children have received the same vaccines multiple times. This is largely due to widespread campaigns (eg, for polio or measles), during which HSAs often go door-to-door and deliver vaccines to all children regardless of whether or not their health cards show that they already received the vaccine.

[During the campaigns], some caregivers are welcoming but some refuse saying that we all know that we receive vaccines at the facility so why have we decided to go to their homes to deliver the vaccines. So they find this unusual and most say then there is something we have put in the vaccines so most of them feel like we are forcing them to [take the vaccines].—HSA from Mzimba North

In addition, children in Lilongwe might also receive repeat vaccines if their health cards are lost; due to the lack of vaccination registers, HSAs in Lilongwe reported their practice is to start the vaccination sequence over again unless the caregiver can confirm without a doubt which doses have already been administered. Some caregivers shared that they dislike seeing their children receive the same dose repeatedly because they fear it might be harmful or do not want their child to experience side effects unnecessarily.

### Subtheme 2: rumours about the COVID-19 vaccine and other new vaccines
In recent years, multiple new vaccines have been introduced for widespread distribution in Malawi, including vaccines against COVID-19, malaria, cholera and now typhoid, further adding to caregivers' concerns about their children receiving too many vaccines and sparking rumours in the community. Rumours and misinformation about the COVID-19 vaccine seemed to be particularly rampant; some caregivers expressed doubts that COVID-19 was even real, and many reported hearing that the COVID-19 vaccine would cause infertility or satanic signs to appear on the body. The concerns about the COVID-19 vaccine have spilled over into the routine immunisations as well; many caregivers mentioned rumours of the COVID-19 vaccine being mixed into childhood vaccines.

I know that children's vaccines are good and they have been good and I got used to them. But the vaccine that scares me is the COVID-19 vaccine because I hear a lot of rumors that when we go to get our children vaccinated they won't vaccinate her with the children's vaccine but that of COVID-19.—PV caregiver from Mzimba North

The combination of COVID-19 rumours, multiple new vaccines being introduced, numerous recent campaigns and the perception of vaccination being forced on people has created an environment in which some caregivers

worry that the government is using vaccination for nefarious purposes and have lost trust in vaccines and other health services.

> Some women started questioning us because they thought they were being forced [to get vaccines]. And some would downright refuse because they have heard rumors that if they get the vaccine they will not give birth, the vaccine will dry their blood and in two years' time they will die and so many other rumors. And now with the coming of the cholera vaccine, things have become complicated again as they are now saying that the COVID vaccine has been incorporated in the cholera vaccine, and they have now started saying we want to kill their children… When they come for under-five clinic and we tell them their child needs to get immunized they tell us no, that it is better for them to stop coming to the facility altogether because they do not feel safe with the immunization.—HSA from Mzimba North

### Opportunity cost of each trip to the facility, exacerbated by long wait times, stockouts and other missed opportunities

**Subtheme 1: long wait times conflict with other responsibilities**

Most caregivers described having numerous responsibilities beyond vaccination, including housework and caring for other children. Lilongwe caregivers also do business activities such as piecemeal work or selling produce. Mzimba North caregivers are busy with farming work, with some working as tenant farmers for employers who do not give them time off for vaccination. Consequently, each trip to the health facility is associated with an opportunity cost, and caregivers must decide how to prioritise their responsibilities.

> [The challenge for tenant farmers is] their work on the farm since they need to follow their employer's orders because as it is the season for rains now it is hard for them to come for scale since they have to go and plant tobacco.—HSA from Mzimba North

Many participants shared that the vaccination process could be very time-consuming due to long travel times, immunisation sessions not starting on time, and some facilities, especially in Lilongwe, being crowded, resulting in slow service.

> The hospital is not always punctual, they open late and some of us have commitments like small businesses. Sometimes we fail to go to the hospital thinking about how inconvenient it gets when we wait too long… some women like me depend much on business and if we come back from there late we find that we did not make anything that day and that gets us worried.—PV caregiver from Lilongwe

**Subtheme 2: missed opportunities, stockouts and minimising vaccine wastage**

Even if caregivers do invest significant time and effort into a trip to the health facility or outreach session, they are not guaranteed to receive service. Vaccine stockouts are not uncommon and, according to HSAs, became even more frequent during COVID-19 due to supply chain challenges.

> I leave everything at home like house chores just to get my child vaccinated, but it hurt me to get there and hear that vaccines are out of stock.—PV caregiver from Mzimba North

Additional missed opportunities resulted from HSAs trying to balance vaccination coverage with reducing vaccine wastage. Caregivers were sometimes sent home without service if they arrived too late, especially if their children needed any vaccines that expire within 6 hours of being opened, such as the measles vaccine. In addition, during some immunisation sessions, especially outreach sessions in Mzimba North, HSAs may not open multidose vaccine vials, such as BCG, unless enough children are present.

> When it is time for scale, the vaccines should be ready and available at the health facilities. This thing where they can't open the vaccine unless there are more than 10 children makes vaccinations [time-consuming]… We can't be going to the health facility now and then just to check if the children are now more than 10.—PV caregiver from Mzimba North

After encountering these situations, caregivers lose motivation and feel they are wasting time by going to the facility if they are not confident they will receive services. The benefits of vaccination are no longer worth the time and effort required, especially when caregivers have other pressing commitments or are concerned about losing their jobs.

### Limited family support and vaccination burden placed largely on the mother

In general, most caregivers said their family members, including husbands, encouraged them to vaccinate their children. Most also described a supportive community environment around vaccination in which they received encouragement and reassurance from other caregivers in their community. For instance, caregivers reminded each other about vaccination dates, travelled to the facility together, or reassured each other when children experienced side effects.

> As neighbors in the community we encourage each other to go to scale on the day for scale. For some who give up, we ask them why they want to give up and then we try to convince them to not give up but to go get their child vaccinated.—PV caregiver from Mzimba North

While a few caregivers, mostly in Mzimba North, shared that their husbands accompanied them to the facility or even brought the child to the facility when they could not, the burden of bringing the child to and from vaccinations still falls largely on the mother. If mothers are sick or busy

with other responsibilities, such as attending a funeral or caring for an ill family member, there may be no one else who is willing to bring the child to the facility when the vaccinations are due.

> One caregiver came with the child's father which made me very happy that the father would also take part in the vaccination and come for the under-5 [clinic]. [Usually] I only see women coming with their children.—HSA from Lilongwe

This responsibility is especially challenging for caregivers who live far from the facility and have limited transport options and those who live in communities with unreliable outreach services. HSAs shared that outreach sessions might be cancelled due to lack of transport or fuel or during bad weather since many outreach sessions take place outside.

Gender norms around vaccination responsibilities and lack of sufficient family support exacerbate other barriers that caregivers face as well. When caregivers are not supported to bring the child to the facility or to care for the child after vaccination, they perceive that the opportunity cost is even higher. Additionally, if children miss vaccines or fall behind schedule when mothers are sick or busy, then caregivers may drop out due to fear of repercussions for falling behind schedule.

### Key differences between Lilongwe and Mzimba North

While the five themes described above largely applied to caregivers in both Lilongwe and Mzimba North, the study also revealed key differences between the two districts. Differences fell across the categories of the BeSD Model and related to vaccine knowledge and education, social support, facility options for vaccine services, distance to facilities, migrant and tenant communities, wait times, outreach sessions, practices around health cards and vaccine wastage practices (table 3).

### DISCUSSION

Our research generated new insights into caregivers' perceptions of being pressured into vaccination and the resulting fear of punishment that can result in drop-out, as well as into the spillover of COVID-19 vaccine fears into routine immunisations being exacerbated by campaigns and new vaccine introductions. Key barriers and drop-out determinants arose across all categories of the BeSD Model and were consistent with those found in other studies, including poor understanding of the vaccine schedule, unreliable outreach services, migration of caregivers, loss of vaccination cards, lack of family support, missed opportunities, poor access to facilities, negative interactions with health workers, stockouts, rumours and misinformation, conflicting responsibilities, and poorly perceived service quality.[12–19]

The key differences in our findings compared with existing literature were new drop-out determinants relating to caregivers' beliefs and perceptions as well as to

vaccination messaging practices. While the pressure that the government and the health sector put on caregivers to get their children vaccinated can be a source of motivation, it can also be counterproductive; caregivers who have fallen off-schedule fear reprimand or punishment for not abiding by the vaccination 'law' and may decide to not return to the facility. Caregivers are also experiencing new concerns and hesitancy due to COVID-19, repeated widespread campaigns, and the addition of new vaccines to the schedule. Many recent studies have documented that COVID-19 vaccine hesitancy has spilled over into routine immunisations, resulting in caregivers having reduced confidence in childhood vaccines.[1 30] However, our data build on this existing literature and further suggest that at least in the context of Malawi, campaigns and new vaccines being introduced to the country might exacerbate this phenomenon.

The strengths of this study are rooted in the CBPR approach; the caregiver researchers drew from their own vaccination experiences and their community connections to bring community-centred insights to the data collection and analysis processes. The variety of data collection methods using different materials and media helped to engage participants and generate rich, detailed data. Finally, the inclusion of participants from both Lilongwe and Mzimba North generated nuanced and contextual findings. There were several limitations to the study. During recruitment, we had challenges recruiting migrant and tenant participants due to the nature of their situations, and recruitment methods were not consistent between study sites due to differences in vaccination record-keeping systems; these may have introduced selection bias. A polio campaign that began partway through the caregiver data collection may have introduced recall bias and influenced the findings related to perceptions of new vaccine introduction and campaigns. Finally, the message exchanges with several HSAs in Mzimba North were much shorter than the others due to network challenges; this may have resulted in their perspectives not being as well documented.

The research findings have implications on vaccination practices in Malawi and beyond. As the world works to rebound from the COVID-19 pandemic, there is increasing recognition that immunisation systems need to be strengthened at many levels. At the community level, it is critical that communities provide input on how services, especially outreach, could better fit around their busy schedules. Community-level social listening could help to gain better insight into caregivers' concerns on a routine basis and quickly address rumours as they emerge. Finally, there is a need for increased community mobilisation targeting men to address social and gender norms that overburden mothers, and targeting tenant employers to help caregivers manage conflicting responsibilities. At the facility level, a sufficient and motivated workforce is needed so that HSAs have time to provide personalised education to caregivers and are motivated to create a welcoming environment for them. Resources

**Table 3** Key differences between experiences of caregivers in Lilongwe and Mzimba North

| Category | Lilongwe | Mzimba North |
|---|---|---|
| Vaccine knowledge and education | Mixed reports about the vaccine education opportunities offered at health facilities. Health talks were sometimes cancelled due to late opening hours, caregivers arriving late or caregivers being impatient for services. | Caregivers were more likely to have knowledge about what to do if a vaccine is missed or what kind of side effects to expect. Caregivers learnt about vaccines and other aspects of child health via health talks and singing songs. |
| Social support for vaccination | Family support came more often in the form of money for transport. Caregivers were more likely to have support from friends or neighbours compared with Mzimba North. | Family support came in the form of assistance caring for the child after vaccination, accompanying the caregiver to the facility or bringing the child to the facility if the caregiver was unwell and unable to go. |
| Facility options for vaccine services | Caregivers had options to go to different facilities (public and private) based on factors like cost, wait times, service quality, stockouts, etc. However, poorer caregivers were still more likely to go to public facilities or to miss sessions than go to private ones. | Caregivers had limited access to other facilities. As a result, if the facility was closed or had a stockout, or if an outreach session was cancelled, there were limited other vaccination opportunities. |
| Distance to vaccinate | Transport cost associated with distance drove caregivers to choose a closer facility especially if they did not have transport money. | Being far from the facility made the vaccination process more time-consuming and made it harder for caregivers to go to the facility when they were feeling unwell or were busy. |
| Agricultural migrant and tenant communities | Urban areas in Lilongwe do not have seasonal agricultural migrants. | Mzimba North is home to many caregivers who migrate seasonally or who work as tenants on farms. These populations often miss vaccines during the farming season, in part due to employers not allowing caregivers to take time off to go for vaccination. Migrant populations' vaccine records are harder to track since they move between different facilities' catchment areas. |
| Wait times | Many participants in Lilongwe described long wait times. HSAs in Lilongwe felt that caregivers are often impatient and want fast service, especially at static facilities. | Few participants in Mzimba North complained about long wait times or about health workers arriving late. Facilities in Mzimba North generally have much smaller catchment populations. |
| Outreach sessions | Caregivers did not mention many challenges with outreach services. | More caregivers talked about outreach services being inconsistent or cancelled. |
| Lost health card practices | If a health card is lost, caregivers must buy a new one and the child will have to repeat all vaccines unless the caregiver can prove that the child has received them. | Losing vaccination cards was rarely mentioned. If a caregiver loses the card, HSAs can review the vaccination registry books to fill out the new card. If a caregiver could not afford a new card, HSAs recorded the child's vaccines on pieces of paper until the caregiver was able to get a new card. There were no reports of repeated vaccinations resulting from lost health cards. |
| Vaccine wastage practices | Most HSAs said that they followed the practice of opening a vaccine vial (eg, of BCG or measles) even if only one child was present. However, this was not always true at outreach sites. | Both caregivers and HSAs mentioned that BCG and measles vaccines are not opened unless there are a certain number of children, especially at the outreach sites. Caregivers were instead told to go to the static facility or to return in a different week. |

HSA, Health Surveillance Assistant.

shortages, including of vaccine supplies and transport for HSAs to travel to outreach sessions, need to be addressed, and infrastructure needs to be improved, especially for outreach sessions, many of which have no shelter from the elements. At the national level, there may be a need to revisit and standardise certain practices, including vaccine education and messaging, especially when new vaccines are introduced. Practices related to vaccine wastage and how to handle caregivers who are missing a health card or children who have fallen behind schedule need to be more consistent and transparent. Finally, there could be many benefits from implementing a national electronic immunisation registry to track vaccination records.[31]

An important area for future research would be to understand how to better engage secondary caregivers, including husbands and other family members, to be actively involved in the immunisation process. Additionally, there is a need to better understand how migrant communities navigate the vaccination process and how to design services that would meet their needs.

**Acknowledgements** Glitter Ndovi assisted with data collection and analysis as a Caregiver Researcher. Nicole Mbouemboue assisted with coding.

**Contributors** JP, JK, CM and EL contributed to study design and logistics. MC and TC contributed to stakeholder engagement necessary to conduct the study. JK, HK and KN contributed to data collection. JP, JK, HK, KN and EL contributed to data analysis. JP took the lead in writing the manuscript, with consultation from JK. All authors provided critical feedback and helped shape the final manuscript. EL is the guarantor of the manuscript. All authors have read and agreed to the published version of the manuscript.

**Funding** This work was supported by Wellcome Trust grant number 224031/Z/21/Z.

**Competing interests** None declared.

**Patient and public involvement** Patients and/or the public were involved in the design, or conduct, or reporting, or dissemination plans of this research. Refer to the Methods section for further details.

**Patient consent for publication** Not applicable.

**Ethics approval** This study involves human participants and the study was approved by the Institutional Review Board of the Malawi National Health Sciences Research Committee (protocol code 22/04/2893 approved on 10 June 2022). Participants gave informed consent to participate in the study before taking part.

**Provenance and peer review** Not commissioned; externally peer reviewed.

**Data availability statement** Data are available on reasonable request. Deidentified data are available on reasonable request.

**ORCID iD**
Jocelyn Powelson http://orcid.org/0000-0002-9935-9122

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
