## [Reviewer comments · BMJ Open]

ARTICLE DETAILS

TITLE (PROVISIONAL)	Using community-based, participatory qualitative research to identify determinants of routine vaccination dropout for children under 2 in Lilongwe and Mzimba North Districts, Malawi
AUTHORS	Powelson, Jocelyn; Kalepa, Joan; Kachule, Hannah; Nkhonjera, Katie; Matemba, Charles; Chisema, Mike; Chumachapera, Tuweni; Lawrence, Emily

VERSION 1 – REVIEW

REVIEWER	Dadari, Ibrahim University of South Florida
REVIEW RETURNED	13-Nov-2023

GENERAL COMMENTS	Thank you for this important work to improve routine immunization coverage in Malawi. Kindly find my comments to improve the paper below. Overall the paper adds value to immunization programming. The paper organization may benefit from some adjustments to avoid having disjointed information. Also, the study limitations could be improved upon. Introduction • Page 4, Line 10 – statement misleading implying 16 different vaccines rather than 16 doses of various vaccines (6 different vaccines BCG, OPV, IPV, Penta, MCV, Rota). Pls clarify the number of doses and not vaccines. “Malawi’s National Immunization Program recommended that all children receive.• Page 4, lines 28-30: Full immunization is not synonymous with dropout as such a distinction be made in the statement “Also in 2015-16, only 2% of 29 children had received no vaccinations, indicating a country-wide vaccination dropout rate of roughly 30 22%.”• In addition to the mentioned survey immunization coverage data, no reference was made to administrative data and WHO-UNICEF estimates of national immunization coverage for Malawi.• Highlights of vaccination coverage rate decline during the pandemic period for Malawi should be briefly discussed and itemized and where possible for the study areas or subnational areas Methods • Pg 5; Lines 6-7: please explain how the four “caregiver researchers” were selected, and the training content they received concisely.• Pg 5; Lines 10-11: which components of the UNICEF Journey to health and immunization were added to the BeSD framework and why?
---

	 • Pg 5; Lines 20-21: how was dropout used to select health facilities. Was it based on a cut-off or threshold, or was it just arbitrary? It'll be good to clarify how. • Pg 5; Lines 27-28: how was vaccination status determined? Was it using vaccination card or recall or both? Clarify how this was done. Also briefly highlight the reasons why children aged 25-34 months were included in the study? • Pg 5; Lines 29-31: could you add in the background/introduction section the specific characteristics or differences on how childhood vaccination is tracked in Mzimba and Lilongwe highlighting specific nuances. • Pg 5; Lines 36-37: why were the two PVs excluded from the study? Some justification should be provided since when they were recruited their children were partially vaccinated and that should not have had any effect even if they decided to vaccinate their kids immediately which is a good thing to do. • Pg 6; Lines 2-6: this should have come way up not under data collection (see comment above) • Pg 6; Lines 19-22: Was there any criteria for choosing photos from the pool? Please detail this out including why some photos were removed or included in the study. A systematic description of the 29 photos and what they contained will be helpful. • Pg 7; Lines 1-11: A flowchart or diagram could be useful in summarizing the steps taken and for the ease of comprehension by the reader. • Pg 7; Lines 13-14: Confirm which language data was collected in?
--	--

REVIEWER	Jelle, Mohamed UCL
REVIEW RETURNED	20-Nov-2023

GENERAL COMMENTS	Routine Childhood Immunization in Malawi: Community based research using participatory qualitative methods to identify determinants of vaccination dropout for children under 2 in Lilongwe and Mzimba North Districts Comments This article presents findings from Community based research using participatory qualitative methods to identify determinants of vaccination dropout for children under 2 in Lilongwe and Mzimba North Districts. The topic is very important for public health, especially in countries where vaccination dropout for children under 2 is a problem. I have some comments and suggestions on the text: General comments  • Please update the Standards for Reporting Qualitative Research (SRQR). For example, under the Researcher characteristics and reflexivity, I couldn't see reflexivity and educational level of the caregiver researchers. • On several occasions, you have mentioned about the study in Mozambique, which is fine but please make sure that readers can read this study as a standalone without needing to go back to the Mozambique study. For example, in the analysis section, you can't just say that you used a codebook that was developed in a previous study. You need to explain in detail, what was done. The title:
---

	 • The title looks long and confusing. Please consider rephrasing it as: “Using community-based, participatory qualitative research to identify determinants of vaccination dropout for children under 2 in Malawi”. Introduction:  • On page 4, from lines 31-39, you described the barriers that can lead to vaccination dropout in LMIC and in particular, Malawi. Could you add the unique contributions of your research to this existing body of information regarding factors affecting vaccination dropout? Methods Study design  • Could you add a few lines to explain why you used a community-based participatory research (CBPR) approach? • Similarly, could you add a few lines on why you used a combination of WHO’s Behavioral and Social Drivers (BeSD) and components from UNICEF’s immunization journey as framework for data collection and analysis? Data collection Photo-elicitation interviews  • It is not very clear to me how this was used. You mentioned that a mixture of photos including 24 from Mozambique and 5 from Malawi, were used in photo-elicitation interviews. However, for readers who have no previous experience with such technique, this is not enough. Could you explain more how this technique works? Perhaps, give one example and explain how it worked. • It would be helpful to briefly explain what Photovoice is and how it was utilized in the current study to provide context for readers who may not be familiar with the technique. Data analysis  • Please describe how you used BeSD and UNICEF immunization journey during the analysis. • You mentioned that the initial codebook was based on the BeSD constructs and UNICEF immunization journey and that it drew from a codebook developed during a similar study in Mozambique. You also mentioned that the codebook was updated by JP and EL. That is not enough. As I mentioned above. We should be able to understand what analysis was done without going back to that other study. Please give more details what was done. • Add reflexivity of the caregiver researchers their level of education, what the training that was provided to them covered etc. Results:  • The results are not very clear. I got confused with the many tables. This is a qualitative study with purposive sampling. Please consider removing some of the tables. I’m not sure how useful table 3 is. • Please try to use a figure (if possible) to summarize the findings. Discussions  • Please begin the discussion by summarizing the key findings of the study concisely. This will provide readers with a quick overview before delving into the detailed analysis.
--	---

	 • While the study references other research studies on page 16 (lines 6-19), consider expanding on the comparisons and contrasts with these studies. Discuss how your findings contribute to or differ from the existing literature, providing a more nuanced understanding of the challenges faced by caregivers in Malawi. • Please expand on the limitations. What is provided so far looks too brief and rushed
--	--

VERSION 1 – AUTHOR RESPONSE

Reviewer: 1 - Dr. Ibrahim Dadari, University of South Florida, United Nations Children's Fund

Comments to the Author:

Thank you for this important work to improve routine immunization coverage in Malawi. Kindly find my comments to improve the paper below. Overall the paper adds value to immunization programming. The paper organization may benefit from some adjustments to avoid having disjointed information. Also, the study limitations could be improved upon.

1. Page 4, Line 10 – statement misleading implying 16 different vaccines rather than 16 doses of various vaccines (6 different vaccines BCG, OPV, IPV, Penta, MCV, Rota). Pls clarify the number of doses and not vaccines. “Malawi’s National Immunization Program recommended that all children receive.

Response: We adjusted the language to indicate that there are 16 doses of vaccines rather than 16 vaccines.

2. Page 4, lines 28-30: Full immunization is not synonymous with dropout as such a distinction be made in the statement “Also in 2015-16, only 2% of 29 children had received no vaccinations, indicating a country-wide vaccination dropout rate of roughly 30 22%.”

Response: We updated the data in this section (see response to the comment below) and have removed this reference to dropout rates.

3. In addition to the mentioned survey immunization coverage data, no reference was made to administrative data and WHO-UNICEF estimates of national immunization coverage for Malawi.

Response: We removed most references to the mentioned survey coverage data and instead replaced it with descriptions of the WHO-UNICEF estimates. We felt that it would be less confusing to cite just one source when describing the coverage trends. The survey coverage data is still cited in reference to the difference between urban vs rural coverage as the WHO-UNICEF estimates are not disaggregated at that level. The text now reads, “Addressing barriers to full vaccination is especially important in Malawi, where WHO and UNICEF estimates of coverage have dropped during the last decade for most vaccine doses; for example coverage estimates for BCG, DPT-HepB-Hib 3, OPV 3, and PCV 3 were all 95% or higher in 2012, but then fell to 86% or lower by 2016. While routine vaccination coverage increased between 2016 and 2019, coverage for almost all doses on the routine schedule declined by approximately 15 percentage points between 2019 and 2022, in large part due to the COVID-19 pandemic. Coverages for most doses were between 82% and 89% in 2022; however, measles second dose coverage continues to lag behind, at an estimated 60% in 2022. Contrary to similar settings in Sub-Saharan Africa, Malawi’s coverage rates are higher in rural areas than in urban areas, at 77% and 70% full vaccination coverage respectively in 2015-16.”

4. Highlights of vaccination coverage rate decline during the pandemic period for Malawi should be briefly discussed and itemized and where possible for the study areas or subnational areas

Response: We highlighted the overall trend in vaccination coverage rates from 2019 to 2022. See previous response for the new text.

5. Pg 5; Lines 6-7: please explain how the four “caregiver researchers” were selected, and the training content they received concisely.

Response: We shifted the paragraph explaining how the caregiver researchers were selected and trained from the “Data Collection” section up to the “Study Design” section where the caregiver researchers are first introduced and added more detail about the content of the training.

6. Pg 5; Lines 10-11: which components of the UNICEF Journey to health and immunization were added to the BeSD framework and why?

Response: We added further clarity on the adaptations that were made to the BeSD framework by explaining that. The sentence now reads: “We adapted the World Health Organization’s Behavioral and Social Drivers (BeSD) of vaccination framework by adding components from UNICEF’s immunization journey and breaking down the “practical issues” domain into practical issues during preparation, point-of-service, and after care to enable development of more nuanced themes.”

7. Pg 5; Lines 20-21: how was dropout used to select health facilities. Was it based on a cut-off or threshold, or was it just arbitrary? It’ll be good to clarify how.

Response: We added more details to note that these facilities were selected largely by EPI based on their data in combination with other geographic characteristics. The start of the paragraph now reads, “To select study sites, we worked with the Malawi Expanded Programme on Immunization (EPI) who provided us with a list of health facilities with highest dropout rates. We then worked in coordination with the national and district EPI to purposively select eight of those facilities (four per district) that represented both rural and urban sites and that were fairly accessible.”

8. Pg 5; Lines 27-28: how was vaccination status determined? Was it using vaccination card or recall or both? Clarify how this was done. Also briefly highlight the reasons why children aged 25-34 months were included in the study?

Response: Further clarification on how we determined vaccination status is provided in the subsequent paragraph, which reads, “[Caregiver Researchers] confirmed eligibility and vaccination status through children’s health cards, when available. Caregivers in Lilongwe who were missing health cards were asked to describe which vaccines their children had received and were excluded if they were uncertain of their child’s vaccination status.”

We added this sentence to clarify why children in that age range were included: “Caregivers were eligible if their child was 25-34 months old, an age by which they should have completed the under-2 immunizations while still being early enough to minimize recall bias.”

9. Pg 5; Lines 29-31: could you add in the background/introduction section the specific characteristics or differences on how childhood vaccination is tracked in Mzimba and Lilongwe highlighting specific nuances.

Response: We added more information about vaccination tracking systems in rural vs urban settings. The text reads, “Reviews of reporting forms and anecdotal evidence from the facilities involved in this study indicate that vaccination record-keeping tools and processes also differ between urban and rural facilities; in rural regions such as Mzimba North, facilities track individual children’s records in paper-based vaccination registers, while in urban settings such as Lilongwe, facilities keep daily, paper-based tallies of antigens delivered, but do not keep records for individual children due to the much higher patient numbers as well as the likelihood of caregivers visiting multiple different facilities for vaccination services.”

10. Pg 5; Lines 36-37: why were the two PVs excluded from the study? Some justification should be provided since when they were recruited their children were partially vaccinated and that should not

have had any effect even if they decided to vaccinate their kids immediately which is a good thing to do.

Response: These caregivers were excluded from the study due to indications that they had gone to get the vaccination services because they were afraid of being reprimanded or punished for having not completed the vaccinations on time. We felt that this may result in them not feeling comfortable sharing their true perspectives and experiences during the interviews and that this could introduce bias. We added a sentence to further explain this reasoning, "They were excluded from the study due to concerns that they went to complete vaccination out of fear of being reprimanded and that this fear might limit their comfort to provide genuine, unbiased responses in the interviews."

11. Pg 6; Lines 2-6: this should have come way up not under data collection (see comment above)

Response: We have shifted this paragraph up to the "Study Design" section.

12. Pg 6; Lines 19-22: Was there any criteria for choosing photos from the pool? Please detail this out including why some photos were removed or included in the study. A systematic description of the 29 photos and what they contained will be helpful.

Response: We have added additional context to describe how the photos were selected and why additional photos from Malawi were added. While we have removed the photos themselves from the appendix in response to the Editor's concerns, we have added descriptions of the photos into the appendix.

13. Pg 7; Lines 1-11: A flowchart or diagram could be useful in summarizing the steps taken and for the ease of comprehension by the reader.

Response: We added Figure 2 to more clearly show the data collection processes and participants for each method.

14. Pg 7; Lines 13-14: Confirm which language data was collected in?

Response: We have noted which languages the data was collected in: "HSAs participated in message exchanges and semi-structured interviews in Tumbuka, Chichewa, or English to share their experiences delivering childhood immunizations and their perspectives on vaccination dropout."

Reviewer: 2 - Dr. Mohamed Jelle, UCL

Comments to the Author:

This article presents findings from Community based research using participatory qualitative methods to identify determinants of vaccination dropout for children under 2 in Lilongwe and Mzimba North Districts. The topic is very important for public health, especially in countries where vaccination dropout for children under 2 is a problem.

I have some comments and suggestions on the text:

1. Please update the Standards for Reporting Qualitative Research (SRQR). For example, under the Researcher characteristics and reflexivity, I couldn't see reflexivity and educational level of the caregiver researchers.

Response: We added additional details about the Caregiver Researchers' educational levels and reflexivity. The text now reads, "Caregiver Researchers were recruited from the study districts via an open job posting. Applicants were required to have a child 36 months or younger, and women were prioritized as they most closely represent the study population. Two women in each district who met those requirements were selected as Caregiver Researchers based on their familiarity with the study communities, fluency in local languages and in English and their prior research experience. All four Caregiver Researchers had Bachelor's degrees and basic data collection and research skills and had recently experienced the under-2 vaccination journey with their own children. They participated in a five-day training on qualitative methods and research ethics that covered interview techniques such as how to minimize bias and how to probe for rich, detailed information."

2. On several occasions, you have mentioned about the study in Mozambique, which is fine but please make sure that readers can read this study as a standalone without needing to go back to the Mozambique study. For example, in the analysis section, you can't just say that you used a codebook that was developed in a previous study. You need to explain in detail, what was done.

Response: We have added some additional context about how the previous codebook was developed. The text now reads, "The initial codebook was based on a codebook developed during a similar study on vaccination dropout in Mozambique that used the same adapted BeSD framework; that codebook was developed deductively based on the BeSD constructs and UNICEF immunization journey. The codebook was updated with new codes added inductively by JP and EL following review of the first six caregivers' interview transcripts and was presented to the Caregiver Researchers for additional edits in order to tailor the codebook to the Malawi context."

3. The title looks long and confusing. Please consider rephrasing it as:

"Using community-based, participatory qualitative research to identify determinants of vaccination dropout for children under 2 in Malawi"

Response: We simplified the title to, "Using community-based, participatory qualitative research to identify determinants of routine vaccination dropout for children under 2 in Lilongwe and Mzimba North Districts, Malawi"

4. On page 4, from lines 31-39, you described the barriers that can lead to vaccination dropout in LMIC and in particular, Malawi. Could you add the unique contributions of your research to this existing body of information regarding factors affecting vaccination dropout?

Response: We have not made any edits to this section of the introduction, but we did specifically highlight our unique contributions to the existing body of information in the discussion, which now reads, "Our research generated new insights into caregivers' perceptions of being pressured into vaccination and the resulting fear of punishment that can result in dropout, as well as into the spillover of COVID-19 vaccine fears into routine immunizations being exacerbated by campaigns and new vaccine introductions... The key differences in our findings compared to existing literature were new dropout determinants relating to caregivers' beliefs and perceptions as well as to vaccination messaging practices."

5. Could you add a few lines to explain why you used a community-based participatory research (CBPR) approach?

Response: We described CBPR in more detail and why we used it. The text now reads, "We conducted a qualitative study using a community-based participatory research (CBPR) approach. CBPR is an approach which engages community representatives throughout the research process, helping to minimize bias and reduce power dynamics during data collection and facilitating development of contextually-sensitive and community-centered findings. We hired and trained four caregivers from the study communities (hereafter referred to as "Caregiver Researchers") to lead data collection and assist with analysis."

6. Similarly, could you add a few lines on why you used a combination of WHO's Behavioral and Social Drivers (BeSD) and components from UNICEF's immunization journey as framework for data collection and analysis?

Response: We added more detail about how we adapted the BeSD framework with components of UNICEF's immunization journey and about why we used the BeSD framework. The text now reads, "We adapted the World Health Organization's Behavioral and Social Drivers (BeSD) of vaccination framework, which is an established framework for qualitative research on caregivers' immunization experiences. We added components from UNICEF's immunization journey by breaking down the "practical issues" domain into practical issues during preparation, point-of-service, and after care to enable development of more nuanced themes. This model served as the framework to guide data collection and analysis (Appendix B)."

7. Photo-elicitation interviews: It is not very clear to me how this was used. You mentioned that a mixture of photos including 24 from Mozambique and 5 from Malawi, were used in photo-elicitation interviews. However, for readers who have no previous experience with such technique, this is not enough. Could you explain more how this technique works? Perhaps, give one example and explain how it worked.

It would be helpful to briefly explain what Photovoice is and how it was utilized in the current study to provide context for readers who may not be familiar with the technique.

Response: We have added more detail about Photovoice and photo-elicitation methods and how they were used. The text now reads, "Photo-elicitation interviews were conducted using semi-structured guides based on the BeSD framework (Appendix C). Photo-elicitation is a method through which photos are incorporated into the interview process, allowing participants to engage with and react to visual material to generate rich, detailed findings(29); for example, participants might be shown a set of photos and then asked to describe if and how those photos relate to their own lives. These methods built upon a sister study on vaccination dropout in Mozambique that employed Photovoice methods, through which caregivers were provided with cameras and instructed to take photos of anything that represented their experiences vaccinating their children, resulting in a set of 159 photos of different aspects of the immunization journey taken between October and November 2020(13)."

8. Please describe how you used BeSD and UNICEF immunization journey during the analysis.

Response: We have added additional detail into the Study Design section to describe how the BeSD framework was adapted with components of the UNICEF immunization journey. In the data analysis section, we have added more detail about how the adapted BeSD framework was used in the analysis as a way to organize barriers and facilitators of complete vaccination within the categories of the BeSD framework so that we could then identify themes that fell across different categories. The text now reads: "Caregiver Researchers participated in a one-week participatory analysis process facilitated by JP, in which they reviewed code summaries and representative quotations from the caregiver interviews, discussed key facilitators and barriers to full vaccination within each category of the adapted BeSD framework, identified themes and patterns of dropout that fell across the BeSD framework categories, and drew comparisons between PV and FV caregivers and between Lilongwe and Mzimba North."

9. You mentioned that the initial codebook was based on the BeSD constructs and UNICEF immunization journey and that it drew from a codebook developed during a similar study in Mozambique. You also mentioned that the codebook was updated by JP and EL. That is not enough. As I mentioned above. We should be able to understand what analysis was done without going back to that other study. Please give more details what was done.

Response: We added more details about how the original codebook was developed as well as about how it was adapted by JP and EL. The text now reads, "The initial codebook was based on a codebook developed during a similar study on vaccination dropout in Mozambique that used the same adapted BeSD framework; that codebook was developed deductively based on the BeSD constructs and UNICEF immunization journey. The codebook was updated with new codes added inductively by JP and EL following review of the first six caregivers' interview transcripts and was presented to the Caregiver Researchers for additional edits in order to tailor the codebook to the Malawi context."

10. Add reflexivity of the caregiver researchers their level of education, what the training that was provided to them covered etc.

Response: We have noted that the caregiver researchers all had Bachelor's degrees and basic data collection experience. We added more detail about the topics covered in the training. The text now reads, "Caregiver Researchers were recruited from the study districts via an open job posting. Applicants were required to have a child 36 months or younger, and women were prioritized as they

most closely represent the study population. Two women in each district who met those requirements were selected as Caregiver Researchers based on their familiarity with the study communities, fluency in local languages and in English and their prior research experience. All four Caregiver Researchers had Bachelor's degrees and basic data collection and research skills and had recently experienced the under-2 vaccination journey with their own children. They participated in a five-day training on qualitative methods and research ethics that covered interview techniques such as how to minimize bias and how to probe for rich, detailed information."

11. The results are not very clear. I got confused with the many tables. This is a qualitative study with purposive sampling. Please consider removing some of the tables. I'm not sure how useful table 3 is.
Response: We removed table 3 as well as the accompanying text on photo-elicitation findings to reduce potential confusion. Appendix D contains descriptions of the photos for readers who are interested in learning more about them. We feel that tables 1 and 2 are necessary for describing the participants and providing more information about which kinds of populations the findings can be generalized to. We have done our best to have the results organized according to the 5 main themes that arose and have tried to more clearly label the sub-themes, where relevant, to reduce potential confusion.

12. Please try to use a figure (if possible) to summarize the findings.
Response: Unfortunately, we were not able to add an additional figure to summarize findings due to reaching the table / figure limit of the journal.

13. Please begin the discussion by summarizing the key findings of the study concisely. This will provide readers with a quick overview before delving into the detailed analysis.
Response: We began the discussion by briefly summarizing the key new learnings that arose from this research. The beginning of the discussion now reads, "Our research generated new insights into caregivers' perceptions of vaccination being forced upon them and the resulting fear of punishment that can result in dropout, as well as into the spillover of COVID-19 vaccine fears into routine immunizations being exacerbated by campaigns and new vaccine introductions."

14. While the study references other research studies on page 16 (lines 6-19), consider expanding on the comparisons and contrasts with these studies. Discuss how your findings contribute to or differ from the existing literature, providing a more nuanced understanding of the challenges faced by caregivers in Malawi.

Response: We have added more detail to describe the key new findings and how they differ from existing literature. The relevant paragraph from the discussion now reads: "The key differences in our findings compared to existing literature were new dropout determinants relating to caregivers' beliefs and perceptions as well as to vaccination messaging practices. While the pressure that the government and the health sector put on caregivers to get their children vaccinated can be a source of motivation, it can also be counterproductive; caregivers who have fallen off-schedule fear reprimand or punishment for not abiding by the vaccination "law" and may decide to not return to the facility. Caregivers are also experiencing new concerns and hesitancy due to COVID-19, repeated widespread campaigns, and the addition of new vaccines to the schedule. Many recent studies have documented that COVID-19 vaccine hesitancy has spilled over into routine immunizations, resulting in caregivers having reduced confidence in childhood vaccines. However, our data build upon this existing literature and further suggest that, at least in the context of Malawi, campaigns and new vaccines being introduced to the country might exacerbate this phenomenon."

15. Please expand on the limitations. What is provided so far looks too brief and rushed
Response: The study limitations have been fleshed out further. The text now reads, "There were several limitations to the study. During recruitment, we had challenges recruiting migrant and tenant participants due to the nature of their situations, and recruitment methods were not consistent

between study sites due to differences in vaccination record-keeping systems; these may have introduced selection bias. A polio campaign that began partway through the caregiver data collection may have introduced recall bias and influenced the findings related to perceptions of new vaccine introduction and campaigns. Finally, the message exchanges with several HSAs in Mzimba North were much shorter than the others due to network challenges; this may have resulted in their perspectives not being as well documented.” The “strengths and limitations” bullet points below the abstract have also been updated accordingly.

VERSION 2 – REVIEW

REVIEWER	Dadari, Ibrahim University of South Florida
REVIEW RETURNED	03-Jan-2024

GENERAL COMMENTS	Thank you for addressing the Reviewer's comments, your manuscript is much clearer and stronger now.
---

REVIEWER	Jelle, Mohamed UCL
REVIEW RETURNED	07-Jan-2024

GENERAL COMMENTS	I have no more comments. I'm happy for the paper to be accepted.
--